# A Memory-Efficient Encoding Method for Processing Mixed-Type Data on Machine Learning

**DOI:** 10.3390/e22121391

**Published:** 2020-12-09

**Authors:** Ivan Lopez-Arevalo, Edwin Aldana-Bobadilla, Alejandro Molina-Villegas, Hiram Galeana-Zapién, Victor Muñiz-Sanchez, Saul Gausin-Valle

**Affiliations:** 1Centro de Investigación y de Estudios Avanzados del I.P.N., Unidad Tamaulipas, Victoria 87130, Mexico; hiram.galeana@cinvestav.mx (H.G.-Z.); saul.gausin@gmail.com (S.G.-V.); 2Conacyt-Centro de Investigación y de Estudios Avanzados del I.P.N., Unidad Tamaulipas, Victoria 87130, Mexico; edwyn.aldana@cinvestav.mx; 3Conacyt-Centro de Investigación en Ciencias de Información Geoespacial, Merida 97302, Mexico; amolina@centrogeo.edu.mx; 4Centro de Investigación en Matemáticas, Monterrey 66628, Mexico; victor_m@cimat.mx

**Keywords:** categorical data, data preprocessing, machine learning

## Abstract

The most common machine-learning methods solve supervised and unsupervised problems based on datasets where the problem’s features belong to a numerical space. However, many problems often include data where numerical and categorical data coexist, which represents a challenge to manage them. To transform categorical data into a numeric form, preprocessing tasks are compulsory. Methods such as *one-hot* and *feature-hashing* have been the most widely used encoding approaches at the expense of a significant increase in the dimensionality of the dataset. This effect introduces unexpected challenges to deal with the overabundance of variables and/or noisy data. In this regard, in this paper we propose a novel encoding approach that maps mixed-type data into an *information space* using Shannon’s Theory to model the amount of information contained in the original data. We evaluated our proposal with ten mixed-type datasets from the UCI repository and two datasets representing real-world problems obtaining promising results. For demonstrating the performance of our proposal, this was applied for preparing these datasets for classification, regression, and clustering tasks. We demonstrate that our encoding proposal is remarkably superior to *one-hot* and *feature-hashing* encoding in terms of memory efficiency. Our proposal can preserve the information conveyed by the original data.

## 1. Introduction

The continuous growth of Information Technologies and services have made it possible to store, process and transmit data of different types. This results in large and growing sources of data about natural, industrial and business processes as well as scientific, social and political activities among many others. Furthermore, the collected data will keep growing at an ever-expanding rate with the rollout of disruptive technologies such as the Internet of Things (IoT), in which a plethora of sensing devices will collect and transmit data from their environments to the Internet, or Big Data, where huge amount of data must be manipulated by several tasks at different stages with diverse requirements. In particular, it is forecasted that there will be around 40 billion of connected IoT devices generating about 80ZB of data in 2025 [1]. The large volumes of data stored, for instance, in central cloud entities on the Internet turns out to be of value if meaningful information could be extracted for producing value-added services and supporting knowledge-based decision-making systems. In this regard, a remarkable interest in data-mining methods has arisen in recent decades, in which case the goal is to examine datasets with diverse properties to efficiently extract patterns describing the behavior of a study phenomenon [2,3,4]. Such data-mining solutions face several challenges mainly because of the particular characteristics of modern real-world datasets, which can be very large in volume of data, and often with different data types in terms of both numerical and non-numerical data. To gain insight into large datasets, data-mining solutions rely on computational models and methods to mimic the ways by which human make decisions relying on patterns and inferences derived from analyzed data [5,6,7]. These methods are known as Machine Learning (ML), which encompasses a variety of scientific disciplines such as Statistics [8], Learning Theory [9], and Computer Science [10]. Due to the huge amount of available data, an important task in ML is to select an appropriate dataset that represents the experience and knowledge about the problem to be learned and solved by a computer system. This task implies to determine the *d* most relevant features of such a problem (feature selection [11,12]) and to define an appropriate sample of *n*
*d*-tuples containing values for those selected features [13,14]. Since computers work on a numerical representation of operations and data, most ML algorithms require that these tuples be encoded as a sequence of numerical values. In this context, a *d*-tuple is represented as a vector of the form $\vec{x}=\left[x_{1},x_{2},\cdots ,x_{d}\right]\in R^{d}$ where $x_{i}$ corresponds to the value of the $i^{\mathrm{th}}$ feature. Formally speaking, a dataset X is generally composed by a set of *N*
*d*-tuples. In many situations, though, a feature $x_{i}$ can be numerically valued, its value cannot be considered strictly a number because it represents an ordered or unordered category. When this occurs, it said that $x_{i}$ is a *categorical feature*. A dataset X containing *d*-tuples with categorical and numerical features is commonly called a *mixed-type dataset*.

Most of the existing ML methods work under the assumption that the dataset’s features belong to a numerical space, in which an order and a meaningful distance exists. On such assumption, ML methods can compute consistently arithmetic operations, central tendency, dispersion and distance measures. Categorical features do not fully satisfy the aforementioned assumption and, therefore, are not amenable to several mathematical operations and calculations, as detailed in Section 2.1. This imposes preprocessing tasks that attempt to transform categorical features into values amenable to numerical treatment. In this attempt different methods have been proposed, some of which are presented in what follows.

### 1.1. Related Work

When a dataset consists only of categorical features, several approaches attempt to extend the notions of distance, centrality and dispersion to non-numerical features relying on their intrinsic constraints. For example, in clustering problems, *K*-modes algorithm [15,16,17,18] uses a dissimilarity measure to deal with the distance between non-numerical *d*-tuples, replacing the means with modes as centrality measure, and using a frequency-based method to update iteratively the cluster centers to minimize a cost function in a similar fashion to *K*-means. ROCK [19] is another approach that computes distances using the Jaccard coefficient [20] from which, and using a threshold parameter, it is possible to determine the neighbors of any non-numerical *d*-tuple. COOLCAT [21] does not rely in arbitrary notion of distance and instead is based on the notion of entropy. COOLCAT aims to find the clusters that minimize the overall expected entropy.

When the dataset is made up of mixed features, numerical and non-numerical, preprocessing tasks are typically carried out to make non-numerical features amenable to numerical analysis. For example, in regression models [22], neural networks [23], or cluster analysis [24], such a preprocessing consists of encoding categorical features into a set of artificial numerical values (*dummy variables*) that take binary values to indicate the presence or absence of some category [25]. Specifically, categorical features are encoded using a *one-hot encoding* scheme, which creates a binary column for each category as it is shown in Table 1.

Other methods such as *deviation coding*, *orthogonal polynomial coding* and *Helmert coding* [26], use more values than just zero and one.

An important concern about these methods is the number of dummy variables. A categorical feature with *w* different values is represented with w−1 dummy variables [27]. In this sense, large values of *w* might produce an overabundance of them that in turn, might pose performance issues (from computational viewpoint) and, worse yet, side effects such as multicollinearity [28,29]. Attempting to deal with this fact, methods such as *feature-hashing* are widely used [30]. In this case, the original feature values are mapped to a smaller set of hash integer values, which in turn, are converted to new feature values via modular arithmetic. However, some of the original values can be mapped into the same hash value (collision), causing noisy data which can produce inconsistent results. Other methods encode non-numerical to numerical values based on the result data (supervised problems) that have proven to be suitable for the situation where the non-numeric feature has many possible values or even when new values appear after the training stage [31,32,33]. In scenarios where the outcome is unknown (unsupervised learning problems), other approaches have arisen. For instance, in [34], authors describe a method to cluster objects represented by mixed features using a similarity measure first proposed for genetic taxonomy problems. The main idea is to cover both types of data with the same framework considering greater weights to uncommon feature values that match. In [35], authors present some variants of *k*-means algorithm that remove the restriction of processing only numerical features. In [36] is presented the *k*-prototypes algorithm, where the similarity between objects is based on a function of both numerical and non-numerical attributes. In this regard, a new two-terms distance function is proposed. The first term corresponds to the usual square Euclidian distance, whereas the second part is a weighted similarity measure on categorical attributes based on one hyper-parameter and the number of mismatches between an object and a cluster prototype. The hyper-parameter allows a complete control on how much the non-numerical attributes should be taken into account for the final clustering. Using a zero value of the hyper-parameter the algorithm behaves exactly as *k*-means without considering non-numerical features. A similar approach is presented in [37], but the authors based their algorithm on the concept of evidence accumulation. The idea of the evidence accumulation is to combine the results of multiple clusterings into a single data partition, by taking the co-occurrences of pattern pairs (with non-numerical values) in the same cluster as the votes for that association. The data partitions are mapped into a similarity matrix of patterns and then, a Spectral Clustering method is applied. In light of the above ideas, some knowledge discovery systems start to incorporate algorithms to deal with mixed types as part of database mining systems and knowledge discovery systems [38]. Another interesting idea, reported in [39], deals with the inclusion of non-numerical variables to Self-Organizing Maps (SOM). In the original SOM, proposed by Kohonen, non-numerical data cannot be handled straightly due to the lack of a direct representation and computation scheme of the distance. Thus, an improvement of the ability of SOM to process mixed data is reported. The main idea is to build a hierarchy graph of the non-numerical values and then measure the distances of the objects according to their distances in the hierarchy.

Given that some ML methods are based on similarities (or disimilarities) measures between pairs of data objects, Gower ([40,41]) proposed a distance measure for data with mixed variables, which includes numeric, binary, categorical and ordinal. Given two data objects ${\vec{x}}_{i}$ and ${\vec{x}}_{j}$, Gower’s distance is defined as $dist\left(i,j\right)=\frac{\sum_{k}\delta_{ijk}d_{ijk}}{\sum_{k}\delta_{ijk}}\in \left[0,1\right]$, where $dist_{ijk}$ is the contribution of the *k*th variable to the disimilarity between ${\vec{x}}_{i}$ and ${\vec{x}}_{j}$, and is computed according to the type of the variable. For binary or categorical (nominal) variables, $dist_{ijk}=1$ if $x_{ik}\neq x_{jk}$ and 0 if $x_{ik}=x_{jk}$. Numeric variables are considered to be interval-scaled variables, and $dist_{ijk}=\frac{|x_{ik}-x_{jk}|}{R_{k}}$, where $R_{k}$ is the range of the *k*th variable. Ordinal (or categorical ordinal) variables are considered to be numeric variables after been transformed by $z_{ik}=\frac{r_{ik}-1}{M_{k}-1}$, where $r_{ik}$ is the position rank index and $M_{k}$ is the highest rank for variable *k*. In its standard form, the weights $\delta_{ijk}$ of Gower’s distance are set to 1 if both measurements $x_{ik}$ and $x_{jk}$ are non-missing, meaning that both objects can be compared on variable *k*, otherwise, comparison is not possible and $\delta_{ijk}=0$; however, those weights can have different values related to the importance of variable *k* or can be regarded as a function of the result of the variables being compared ([40]). By applying Gower’s distance formula to all pairs of *n* observations (data objects), we get a n×n symmetric, positive semi-definite disimilarity matrix D, which can be used as an input to distance-matrix-based non-supervised ML algorithms, such as hierarchical clustering; but in the case of algorithms based on data features for supervised and non-supervised ML, it is necessary to find some mapping from disimilarities (which encodes the mix of variables) to a representation amenable to those algorithms. The traditional approach is to use Multidimensional Scaling (MDS, [42]), which is a set of algorithms to find representations of distances among points (data objects) in a low-dimensional space, starting from the disimilarity matrix by optimizing a convenient stress function ([43,44,45]) in such a way that the distances between points in the resulting configuration correspond closely to the disimilarities given in D. In its simplest form, classical MDS is equivalent to apply Principal Component Analysis (PCA) to the disimilarity matrix, and the mapping is given by the first *p* principal components.

Similar to Gower, Gibert et al., [46] proposed distance metrics for non-homogeneous data which takes into account quantitative (numeric) and qualitative (categorical) variables indexed by the sets C and Q, respectively. In this case, the distance between data objects ${\vec{x}}_{i}$ and ${\vec{x}}_{j}$ is given by the so-called mixed function: $d_{\left(\alpha ,\beta \right)}^{2}\left({\vec{x}}_{i},{\vec{x}}_{j}\right)=\alpha d_{C}^{2}\left({\vec{x}}_{i},{\vec{x}}_{j}\right)+\beta d_{Q}^{2}\left({\vec{x}}_{i},{\vec{x}}_{j}\right)$, where the distance for quantitative variables is defined by $d_{C}^{2}\left({\vec{x}}_{i},{\vec{x}}_{j}\right))=\sum_{k\in C}\frac{{\left({\vec{x}}_{ik}-{\vec{x}}_{jk}\right)}^{2}}{s_{k}^{2}}$, which is the Euclidean distance normalized by the variance $s_{k}^{2}$, and, for qualitative variables, the distance is defined by $d_{Q}^{2}\left({\vec{x}}_{i},{\vec{x}}_{j}\right)=\frac{1}{n_{Q}^{2}}\sum_{k\in Q}d_{k}^{2}\left({\vec{x}}_{i},{\vec{x}}_{j}\right)$, where $d_{k}^{2}\left({\vec{x}}_{i},{\vec{x}}_{j}\right)$ is the $\chi^{2}$ distance where a symbolic representation of categorical values is used (see [46] for details). The weights α,β≥0, represent the influence given by each group of variables, and some heuristic for choosing these values are proposed in [46]. An extensive sets of experiments for clustering synthetic and real data sets were presented in [46,47], where a hierarchical clustering algorithm is used with the distance metric proposed by Gilbert et al., showing good properties compared with other metrics.

We have mentioned several methods that map from the categorical space to a numerical space in which arithmetic operations and distance measures are meaningful. But these methods may not be as efficient in terms of computational costs and data consistency as supposed. For instance, *one-hot encoding* has the drawback that it increases the dimensionality of the dataset in function to the cardinality of the categorical values. Methods such as *feature-hashing* can be much lower than *one-hot encoding*, though two or more categories can collide when they map into the same hash value, inducing a possible inconsistency in the data. In the case of Gower and Gilbert distances, although it can be very useful when it is used with MDS, it is very limited to small or medium-size datasets, because the solution is given in terms of the number of observations. Those mentioned issues are critical in huge datasets (with hundreds of categorical variables with hundreds of values), such as those for social science, e-commerce, among others, in terms of memory and efficiency because the patterns that we can find in low-dimensional spaces are lost in higher dimensions (*the curse of dimensionality*). Under these drawbacks, we propose a method that maps the original data (including numerical and categorical features) into a new information-theoretic space wherein all data are amenable to numerical analysis.

### 1.2. Contributions

In the attempt to make categorical features amenable to numerical analysis, most of the existing approaches involve the following drawbacks: (1) overabundance of variables leading to increased computational cost in terms of space and time, (2) noisy data as results of encoding processes, and (3) non-standard similarity notions that prevent quantitative comparisons and statistical estimations. Based on these drawbacks and taking into account the need to handle categorical data in ML tasks, we note the following contributions of our work:In contrast with other methods, our proposal does not induce an overabundance of dummy features, minimizing adverse side effects such as multicollinearity and poor performance of ML models in terms of computational costs (space and time).Although methods such as *feature-hashing* reduce the overabundance of features, this involves an important parameter related to the resulting number of hashes which must be decided beforehand. This parameter can cause some original values to be mapped into the same hash value (collision), causing spurious data which can produce inconsistent results. In this regard, our proposal achieves a trade-off between computational costs and data consistency.We propose an encoding method that maps mixed-type data to a metric space in which the data are amenable to arithmetic operations and standard distances, enabling the inherent numerical analysis in the most common ML algorithms regardless of belonging to a supervised or unsupervised approach.

The presentation and discussion of our proposal is organized as follows. Section 2 briefly presents key background concepts. In Section 3 we detail the proposed encoding approach. Afterwards, experiments and the corresponding results are depicted in Section 4. Finally, the main conclusions are given in Section 5.

## 2. Background

In this section, we present what we consider to be the most important elements to discuss and describe our proposal.

### 2.1. Type of Features

We have pointed out that though a feature can be numerically valued, its value cannot be considered strictly a number. There are *nominal features* whose values represent categorical descriptions—with no intrinsic ordering—that induce a partition into mutually exclusive classes. Some examples of these type of features are *nationality*, *race*, *sex* among others. When the values of a feature maintain a categorical description with an intrinsic order, this is called *ordinal feature*. For example, age group is an ordinal feature because there is a meaningful ordering from the lowest to the highest categorical descriptions (e.g., 1 = newborn, 2 = infant, 3 = toddler, 4 = preschool, 5 = school-age child, 6 = adolescent). Since the numerical values can be assigned in a discretionary way, the distances between them could be no meaningful. A feature whose values induce an order as well as a meaningful distance between them is called an *interval feature*. A typical example is the temperature whose values maintain an order ($10{\;}^{{}^{\circ}}C<11{\;}^{{}^{\circ}}C<12{\;}^{{}^{\circ}}C$); moreover, the separation interval between the adjacent values is the same, which allows definition of a coherent distance between either value. A common inconvenience of this feature is the lack of an absolute zero value to denote the lowest possible value. For example, the temperature value 0 ${}^{{}^{\circ}}$C does not mean a lack of hot or cold, it is simply a value. This fact disables operations such as multiplication. For instance, $0{\;}^{{}^{\circ}}C*5{\;}^{{}^{\circ}}C$ or $2{\;}^{{}^{\circ}}C*10{\;}^{{}^{\circ}}C$ are meaningless when talking about the temperature feature. A feature that overcomes this drawback is called *ratio feature*. The presence of an absolute zero makes the multiplication and division (ratio) meaningful. Furthermore, due to the existence of a zero value, this feature does not have negative values. Examples of ratio variables include height, weight, number of clients, among others.

In Table 2 a summary of the properties of the mentioned types is shown. Nominal and ordinal features do not fully present these properties, preventing the calculation of arithmetic operations and statistics. Since these operations and statistics are usually inherent in the most common ML algorithms, it is compulsory that nominal and ordinal features are transformed in values amenable to them.

### 2.2. Information Content

Although a categorical feature is not strictly amenable to numerical analysis, it can be considered a random variable *Y* on a sample space Ω representing all possible levels or categories of it. For instance, given a categorical feature representing the gender of a person, we can define a discrete random variable *Y* on Ω={male,female}, as follows:(1)$$Y\left(\omega \right)=\left\{\begin{array}{cc} 1 & \mathrm{if}\;\omega =\mathrm{male}\\ 2 & \mathrm{if}\;\omega =\mathrm{female} \end{array}\right.$$
We can appeal to Information Theory to measure the information conveyed by each value of *Y* denoted as $y_{i}$. This information is denoted and expressed as [48]:(2)$$I\left(y_{i}\right)=\log\left(\frac{1}{p(y_{i})}\right)=-\log\left(p\left(y_{i}\right)\right)$$
The log function may be taken as $\log_{2}$, in which case, the information is expressed in bits, otherwise, this is expressed in nats (in our approach, we use $\log_{2}$). Notice that $I(y_{i})$ is inversely proportional to the likelihood of $y_{i}$. Later, we show that such a measure is itself a numerical value that can be used as the resulting transformation of $y_{i}$. The expected value of *I* is commonly known as *entropy* and represents the level of uncertainty inherent in the values of *Y*. This value is expressed as:(3)$$H\left(Y\right)=-\sum_{y_{i}\in \Omega }p\left(y_{i}\right)\log\left(p\left(y_{i}\right)\right)$$

### 2.3. Evaluation Metrics for Machine Learning

The encoding of categorical features has an important effect in the performance of ML algorithms. Supervised algorithms find a function *f* that maps a *d*-tuple $\vec{x}$ to a response value *y* based on a set of $\vec{x}-y$ pairs known as a *labeled dataset*. When *y* is a categorical value, we are facing a classification problem, otherwise, we are facing a regression problem. For classification, the performance is commonly determined via a set of metrics based on the so-called confusion matrix. This is an M×M matrix, where *M* is the number of classes being predicted. Each row of the matrix represents the instances in a predicted class while each column represents the instances in an actual class (or vice versa). From this matrix, metrics such as *Accuracy* (the proportion of the total number of predictions that were correct), *Precision* (the proportion of positive cases that were correctly identified), *Sensitivity* or *Recall* (the proportion of actual positive cases which are correctly identified) and *Specificity* (the proportion of actual negative cases which are correctly identified) can be calculated. For regression, the metrics *Root Mean Squared Error*, *Root Mean Squared Logarithmic Error*, *R-Squared*, and *Adjusted R-Squared* are widely known; the interested reader can see [49,50]. The above metrics are based on a reference value *y* that induces an error measure or loss function. This is not the case for unsupervised algorithms where the output value *y* is lacking. Here the algorithm must find a suitable partition on the set of *d*-tuples exclusively guided by proximity criteria based on similarity measures (e.g., Euclidean, Mahalanobis, Manhattan, Minkowsky, Cosine, etc.). Since there is no ground truth, the most common evaluation metrics examine the quality of the partition found based on criteria associated with the shape and geometry of the clusters or subsets in such a partition. These metrics are known as *validity indices* [51]. When it is only required to evaluate the ability of an unsupervised algorithm to find a target partition, a labeled dataset is often used. In this case the evaluation is based on a score function indicating the difference between the output *y* and the cluster label $\hat{y}$ inferred by the algorithm. In this regard, score functions as *Rand Index*, *Adjusted Rand Index* (ARI) [52] and *Mutual Information* [53] are often used.

As mentioned, the encoding of categorical data has a significant impact on the performance of ML algorithms. We quantify this impact by means of *Accuracy* for classification, *Mean Square Error* for regression, and ARI for clustering (see Section 4).

## 3. Information-Based Encoding Method

We start by providing the necessary notation that will be used throughout the description of our proposal. Let X be a mixed-type dataset containing *N*
*d*-tuples of the form $\left(x_{i1},x_{i2},\cdots ,x_{id}\right)$ representing a feature vector that contains numerical and categorical values. These vectors can be arranged in a N×d matrix of the form:(4)$$X=\left[\begin{array}{cccc} x_{11} & x_{12} & \cdots & x_{1d}\\ x_{21} & x_{22} & \cdots & x_{2d}\\ \vdots & \vdots & \ddots & \vdots \\ x_{N1} & x_{N2} & \cdots & x_{Nd} \end{array}\right]$$

Let X[,j] be the $j^{th}$ column of X containing the values of the $j^{th}$ feature (with j=1,2,3⋯,d). Without loss of generality, we assume that numerical features in X are valued in R, and categorical features are valued as non-empty strings that satisfy the regular expression ${\left\{A-Z,a-z,0-9\right\}}^{+}$. Relying on such an assumption, for each X[,j] containing categorical values, we map each unique instance to an integer value. At this point, X will contain numerical values representing both categorical and numerical features.

### 3.1. Preliminary Processing

For each X[,j] representing a numerical feature, we divide its range of values into a set of intervals denoted as Q. Each interval in Q is defined as $q=[l_{q},u_{q})$ where $l_{q}$ and $u_{q}$ represent the lower and upper limits, respectively. For all q∈Q the interval size is given by:(5)$$\delta =\frac{\left|\max(X[,j])-\min(X[,j])\right|}{\eta }$$
where η specifies the total number of intervals. Typically, the value of η is suggested by some rules of thumb such as Sturges’ rule [54], Doane’s formula [55] or the Rice rule [56]. For this work we used Sturges’s rule, which suggests a value of η based on the following expression:(6)$$\eta =\lceil 1+\log_{2}N\rceil$$
Having defined the value of δ, the first interval $q_{1}$ is calculated as a half-closed interval of the form:(7)$$q_{1}=\left[\min\left(X\left[,j\right]\right),\;\min\left(X\left[,j\right]\right)+\delta \right)$$
and the subsequent intervals $q_{2},\cdots ,q_{\eta }$ are defined as follows:(8)$$q_{r}=\left\{\begin{array}{c} \left[u_{q_{r-1}},\;u_{q_{r-1}}+\delta \right]\mathrm{if}\;r\neq \eta ,\\ \left[u_{q_{r-1}},\;u_{q_{r-1}}+\delta \right)\;\mathrm{otherwise}\; \end{array}\right.$$
Each interval is identified by a numerical label corresponding to its order in the set of intervals Q. Then, each numerical value in X[,j] is replaced by the label of the interval that covers this value. At this point, the resulting X is a dataset containing discrete values representing categorical and numerical features. However, so far, X is not yet amenable to arithmetic operations and similarity notions necessary in most ML algorithms. In what follows, we describe an additional process that maps X to a space where this constraint is not present. Such a process is the main core of our proposal.

### 3.2. Information-Based Encoding

As a result of the above processing, the dataset X is composed of *N* discrete *d*-tuples. Each X[,j]∈X can be treated as the sample space of a discrete random variable χ. The probability that χ takes the value $x_{ij}$ (∀$x_{ij}\in X\left[,j\right]$) is denoted as $p(x_{ij})$. This probability can be estimated as $p\left(x_{ij}\right)=\frac{n_{i}}{N}$ where $n_{i}$ is the number of times $x_{ij}$ appears in X[,j] and *N* is the number of *d*-tuples in X. Having determined $p(x_{ij})$, we can now calculate H(χ) representing the expected information conveyed by χ (see Equation (Equation 3)). As a consequence of Gibbs inequality [57], we have that:(9)0≤H(χ)≤log(m)
where *m* is the number of unique values in X[,j]. We have H(χ)=0 when exactly one of the probabilities $p(x_{ij})$ is one and all the rest are zero. On the other hand, we have H(χ)=log(m) only when all probabilities $p(x_{ij})$ are $\frac{1}{m}$, in which case, log(m) corresponds to the maximum possible expected information of χ, denoted for simplicity as $H_{max}$.

The values $x_{ij}$ are defined in a discrete space S, on which, as mentioned, there are constraints regarding either all or some arithmetic operations. Relying on these constraints, we look for a real-valued function T:S→R in which R satisfies a *field structure* for which addition, subtraction, multiplication, and division are defined. In this regard, we define *T* in terms of the contribution of $x_{ij}$ to H(χ) (entropy of X[,j]), as follows:(10)$$T\left(x_{ij}\right)=\left\{\begin{array}{cc} \frac{p\left(x_{ij}\right)I\left(x_{ij}\right)}{H(\chi )} & \mathrm{if}\;H\left(\chi \right)<H_{max}\\ \infty & \mathrm{otherwise} \end{array}\right.$$

The values obtained after applying the above function $\forall x_{ij}\in X\left[,j\right]$ will be the encoded version of X[,j]. Notice that if the condition $H\left(\chi \right)<H_{max}$ is not met, we are facing a feature X[,j] with equiprobable values, in which case, the contribution of each of them is exactly the same and X[,j] would be encoded as a *constant feature*, which as part of an encoded dataset, can lead to errors and provide no information to ML methods. To deal with this issue, we have defined *T* as a piecewise function wherein the second part assigns an infinity value meaning an equitable contribution. The encoded version of X[,j] containing values of *∞* will be not considered to be part of the encoded version of X.

Our method quantifies the contribution of each discrete value $x_{ij}\in X$ relative to the entropy of the feature to which it belongs, denoted as $T(x_{ij})$. Denoting S as a discrete set containing $x_{ij}$ and I as the set containing $T(x_{ij})$$\forall x_{ij}$, our proposal can be expressed as a transformation $S^{d}\to I^{d}$. The resulting dataset is made up of real values (in terms of entropy) that satisfy field properties that make possible arithmetic operations. Notice that the dimensionality of the resulting dataset is not increased during the process. The *N*
*d*-tuples that make up this dataset, induce a *normed vector space* amenable to the notion of proximity in terms of similarity measures.

The idea behind our method is illustrated in Figure 1 by using a small hypothetical dataset. As can be seen, the method is composed of three main stages. In the first stage, *Ordinal Encoding*, the values of all categorical features are mapped to ordinal values (if necessary). In the second stage, *Discretization*, real-valued features are mapped into discrete values representing intervals which are defined according to Equations (Equation 7) and (Equation 8). At this point, we have a dataset containing just discrete values on which the third stage, *Information Content and Transformation*, is applied to obtain real values belonging to what we have called information space. Such a transformation is based on Equation (Equation 10). At this point, we have a dataset containing real-valued features, which can be used in different ML/statistical tasks. Notice that the number of dimensions of the dataset remains invariant to the transformations of each stage. This suggests a better performance in terms of memory when compared to other encoding approaches.

### 3.3. Algorithm

According to the above description, the encoding process of X is summarized by means of Algorithm 1. On the assumption that X includes categorical features valued as non-empty strings, the function ***ordinal_encoding*** is executed, resulting in a new version of these features encoded as numerical ordinal values. Over those real-valued features, the function ***discretization*** is applied. At this point X is a dataset consisting exclusively of discrete numerical values. For each feature X[,j]∈X, the entropy $H_{\chi_{j}}$ is calculated via ***feature_entropy*** likewise the maximum entropy $H_{max_{j}}$ (see Equations (Equation 3) and (Equation 9)). Each value $x_{ij}\in X\left[,j\right]$ is encoded by means of the function ***transformation*** (see Equation (Equation 10)), which requires previously having calculated the information of $x_{ij}$ through the function ***information_content***.

The input of Algorithm 1 is a dataset X composed of *N*
*d*-tuples. Then, we can express the number of operations required by the main processes, as a function of the form τ(d,N). Since typically d≤N, we could define the worst case as the one in which d=N, in which case, τ will grow asymptotically no faster than $N^{2}$ and thus, $\tau \left(d,N\right)\in O\left(N^{2}\right)$.

**Algorithm 1**: Information-based encoding

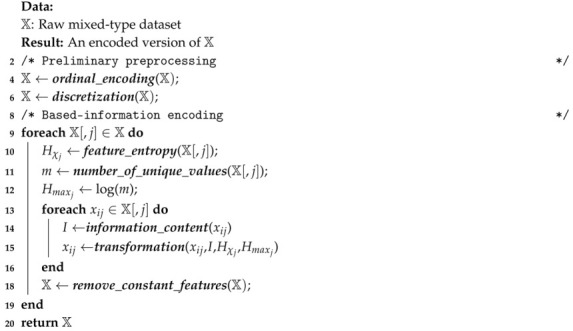



In summary, our proposal maps a mixed-type dataset into an information space in which the data are amenable to arithmetic operations and the proximity notions defined in terms of a metric are meaningful. In what follows, we show that this mapping preserves the information content as well as the most common encoding methods but outperforms them in terms of memory efficiency. Such a preservation is quantified in terms of the effectiveness of the ML algorithms to perform classification, regression, and clustering tasks using the encoded data.

## 4. Experimental Results

This section describes the set of experiments carried out to evaluate the performance of the proposed *information-based encoding* approach and two state-of-the-art encoding approaches (specifically, *one-hot encoding* and *feature-hashing*). In particular, the aim of the presented experiments is two-fold. First, we quantify the performance of the three encoding approaches in terms of the number of features after executing the encoding process, from which we estimate the memory efficiency exhibited by our proposed encoding with respect to the benchmark approaches mentioned. Second, as a step forward, we validate the feasibility of using the developed encoding solution in traditional ML tasks, such as classification, regression, and clustering.

For testing our approach and demonstrating its performance, we required mixed-type datasets, which were obtained from the UCI Machine-Learning Repository. These datasets were chosen on the assumption that was obtained from different real-world problems containing both numerical and categorical values. We wanted to test with few (9) and several (70) attributes. We included two purely categorical datasets for observing the performance of our approach with this characteristic. The main characteristic of these datasets is they have the response variable, which makes them able for prediction (classification and regression) and clustering (as “ground truth” guiding). The properties that were taken into consideration are the number of features, the number of categorical features, and the number of instances per category; the number of classes was not relevant for experiments. The experiments were carried out for demonstrating the transformed datasets can be able for ML/statistical tasks. In Table 3 a brief description of the datasets is given.

Before applying any transformation on these datasets, we complete missing information (usually encoded as blanks, NaNs, or other placeholders) using multivariate imputation [58].

For experimentation, the Algorithm 1 was implemented in Python version 3.8. All the experiments were run on a computer with Intel${}^{®}$ Core™ i7 processor, and 16 GB of RAM.

### 4.1. Memory Efficiency

As first approach to assess the memory efficiency of our proposal, we determined the number of dimensions of the encoded version of the datasets via *one-hot encoding*, *feature-hashing* and our proposal, called in what follows *information-based encoding*; as shown in Figure 2.

A remarkable fact in this case is that *information-based encoding* does not involve an increase in number of dimensions of the encoded dataset. We quantified this increase in percentage terms, where *information-based encoding* exhibits the best value (0%) as shown in Figure 3.

### 4.2. Classification Ability

First, we wanted to quantify the impact of the encoding process on classification tasks using different algorithms. For this, we choose a set of conceptually different classifiers: (1) Decision Trees (DT), (2) Naive Bayes (NB), (3) Multilayer Perceptron (MLP), and (4) Support Vector Machine (SVM) [59,60,61,62]. For each dataset, we obtained three encoded versions via the two benchmark encoders and the *information-based encoding*. The corresponding training and testing sets were obtained for each encoded version of the datasets for each of the chosen classifiers, as it is illustrated in Figure 4. In the testing stage, we make use of unknown instances (not used in the training stage). However, keep in mind that such unknown instances should be statistically similar to those used in the training stage. Furthermore, it is important to note that for both training and testing stages, numerical and categorical values of instances have been mapped to its corresponding interval (the new numerical space *-information space-*) by the *information-based encoding*, "raw" numerical and categorical values are not used.

The classification ability was quantified as the global average of the *true accuracy* of each classifier. This accuracy is estimated by means of a *k*-fold cross-validation with k=10, where the encoded version of the dataset, denoted as $X^{\prime }$, was divided into *k* disjoints subsets $F_{i}$ (i=1,2,⋯,k); being the sets $X^{\prime }-F_{i}$ and $F_{i}$ the training and validation sets respectively. From validation process, a set of *k* accuracy values were obtained and averaged to estimate the true accuracy of the classifier as is shown in Table 4.

Notice that the direct goal of the *information-based encoding* is not to improve the performance exhibited by the classification algorithms (such a performance depends on other elements beyond encoding), instead, the *information-based encoding* aims to preserve the information conveyed by the original dataset. We have quantified such preservation in terms of the accuracy of classification, on the argument that this accuracy is, among other things, a consequence of the ability of the encoding to retain relevant information to discriminate the objects belonging to different classes. The above results show very close accuracy values between *one-hot* and *information-based encoding*, indicating that *information-based encoding* is as good as one of the most popular encoding approaches in terms of information preservation but remarkably superior in terms of memory efficiency (see Figure 3). The worst accuracy was exhibited by *feature-hashing*, perhaps due to a loss of the information content inherent to its previously mentioned drawbacks.

To provide statistical significance regarding the above results, we conducted an analysis of variance (ANOVA) on the accuracy values, regardless of the classifier, to assess if the encoding approaches (covered in this work) have similar effects on the classification ability. In this regard, we stated the hypothesis $H_{0}$ that the differently encoded versions of the data problem induce a similar accuracy. To test this hypothesis, we use the *F-statistic* defined in terms of between-groups and within-groups variances, obtaining in this case, a value of 0.20. Our analysis included three groups (one for each encoding) with 10 samples per group (one for each dataset) from which the degrees of freedom were determined (2 and 7 respectively). In Figure 5 it is shown the region that corresponds to *F*-values greater than or equal to our analysis’ *F*-value (0.20). When $H_{0}$ is true, *F*-values fall in this area with a *p*-value of 0.82. This value represents a quantitative strength of evidence to favor the acceptance of $H_{0}$. The above suggests that all encoding approaches exhibit similar effects in the classification ability. This means that our approach is as good as the others to preserve the information, conveyed by the original dataset, to attain a classification, with the advantage of being more competitive in terms of memory.

### 4.3. Clustering Ability

By clustering ability, we mean the proficiency of the encoding to retain existing relationships between objects (*d-tuples*) from the original dataset. These relationships (typically defined in terms of a notion of similarity) induce a partition or clustering on the data space. To quantify this proficiency, we use the encoded versions of the same datasets but now with the objective of finding a partition as close as possible to the one defined by the response variables. In order to find this partition, we use a set of conceptually different clustering algorithms: *k*-means (KMS), DBSCAN (DBS), and Agglomerative (AGG). The closeness between a partition found by an algorithm and the one induced by the prior labels is defined in terms of the ARI [63]. In summary, for each dataset, we obtained three encoded versions using the benchmark encoders and the *information-based encoding*. Each encoded version is used by the clustering algorithms to find a partition, from which an ARI value is obtained. In Table 5 it is shown the average ARI after 100 executions of the above process.

As in the classification task (see Section 4.2), the objective of the *information-based encoding* is not to improve the performance exhibited by the clustering algorithms (such a performance depends on other elements beyond encoding), instead, we aim to demonstrate that the *information-based encoding* retains existing relationships between objects (*d*-tuples) from the original dataset. To assess this ability, we use the ARI value, on the argument that if *information-based encoding* obtains values of the same order as those obtained by the benchmark methods, it means that the *information-based encoding* can retain the same information as the other encodings. Again, the results show very close ARI values between *one-hot* and *information-based encoding*, indicating that our proposal is as good as one of the most popular encoding approaches in terms of information but remarkably superior in terms of memory efficiency.

We also conducted ANOVA, on ARI values, to test the hypothesis that the differently encoded versions of the data problem, induce a similar clustering performance. We obtained a *F*-value of 0.02. for three groups (one for each encoding) with 10 samples per group (one for each dataset) from which the degrees of freedom were determined (2 and 7 respectively) obtaining a *p*-value of 0.98 that favors the acceptance of $H_{0}$. This result confirms that the *information-based encoding* is as good as *one-hot*, one of the most popular in terms of information, but remarkably superior in terms of memory.

### 4.4. Real-World Cases

For completeness, we provide two real-world cases in which *information-based encoding* was applied. The first one is based on a mixed-type dataset that includes features associated with the diagnosis of COVID-19 in Mexico. The other one is a dataset containing features of software components to estimate the development effort (hours). In Table 6 it is shown the general characteristics of these datasets.

The COVID-19 dataset was collected by health institutions in Mexico and it is composed of a set of patient characteristics associated with the diagnosis of the disease, including demographic information, age, comorbidities, among others. The software component dataset was compiled by a technology company based on its experience in developing software engineering projects. This dataset includes attributes of software components associated with their development effort (hours), among which are complexity (defined according to the qualitative properties of the component), technology (programming languages, frameworks), type (frontend, backend, database, service, etc.), and the time planned by the development team (planned hours).

In this case, the datasets were encoded using *one-hot* and *information-based encoding*. In this step, the memory efficiency in terms of the number of resulting features was calculated for each encoded dataset. As in previous datasets, *information-based encoding* did not increase the number of features, as illustrated in Figure 6.

Considering the encoded versions of COVID-19, we used those classifiers that turned out to be the most prominent, from previous experiments (see Section 4.2), in order to infer a discriminant model. As in these experiments, we also estimate the true accuracy of the model by means of a *k*-folds cross-validation with k=10. The accuracies obtained are shown in Table 7, where the closeness between the encodings again arises. We quantified this closeness as the absolute difference between the accuracy achieved with the encoded version of the dataset through *one-hot* and *information-based encoding*. These results confirm once again that the *information-based encoding* is as good as a *one-hot* in terms of information but extremely superior in terms of memory efficiency.

Since the software component dataset represents a regression problem, we used DT and MLP for inferring a prediction model of a response variable inherently continuous (development effort) as a function of a set of predictors (component properties). Due to the nature of the regression task, in this case, the goodness of such a model was measured in terms of the *Mean Squared Error* (MSE), as shown in Table 8.

### 4.5. Discussion

In what follows, we point out some important remarks relative to our proposal.

Since our proposal is a preprocessing method, regardless of the ML/statistical tasks, it is mandatory to have knowledge from the whole dataset, in order to map the original values into encoded data in terms of information content. The aim is to map categorical and numerical features of a dataset into a numerical space (information space), in which categorical values are amenable to numerical analysis. We resorted to the concepts of *information content* and *entropy* to propose a codification that guarantees the conservation of the information inherent in the original dataset without the need to induce transformations that involve an increase in the memory demand of the encoded data.According to the experiments, we have shown that for ML tasks, our proposal yields comparable performance to benchmark approaches, in terms of the preservation of the information conveyed by the original data, but with a remarkable benefit, in terms of memory efficiency. It is important to remark the experimentation framework was not designed for enhancing the performance of the applied ML algorithms.As a result of the encoding process, we could obtain features with equiprobable values that induce identical encoded values. This situation provides no information to ML algorithms, which can make them prone to errors. For this reason, this kind of feature is removed. We verify that in the space of the encoded data (information space), such removal does not represent an alteration in the ML/statistical tasks.The implementation of the method also allows us to associate each encoded value in the information space to its corresponding value in the original space. This allows us to execute a ML/statistical tasks in the encoded data space, and then, use and interpret the results (inferred class labels, predicted values, or cluster labels) in the original dataset space.

## 5. Conclusions

In this paper, we described a general method, named *information-based encoding*, to encode mixed-type data into a numeric representation denoting the information of each feature and preserving the size of the original dataset. The proposed encoding is based on Information Theory and it can deal well with categorical and numerical values in the original dataset. The main idea behind the proposed approach is to measure how much information is contained in certain intervals of each variable. These intervals are synthetically created from the original distribution of values of each variable for a given dataset. The *information-based encoding* can transform mixed-type data from the features space to an information space (real-valued features), in which the new data are amenable to arithmetic operations and similarity measures. Thus, the encoded dataset can be used by the most common ML algorithms, in both supervised and unsupervised tasks. For demonstrating the goodness of our proposal, this was implemented in Python and, applied to classification (decision trees, naive Bayes, multilayer perceptron, and support vector machine), regression, and clustering (k-means, DBSCAN, and agglomerative) tasks. The obtained global accuracy was quasi-similar to other standard methods (*one-hot* and *feature-hashing*), with the difference that the information-based encoding does not induce overabundance of dummy features (in fact the number of features stay constant) or spurious data (as a consequence of collisions when methods such as *feature-hashing* are used). The experimental results also reveal that our method has the potential for encoding large-scale mixed-type datasets, which are common in different applications and domains such as computer vision, pattern recognition, business intelligence, among others. As a result of encoding, some features may have identical encoded values; this is due to equiprobable values on features during the transformation process. For avoiding biased results on ML/statistical tasks, these features are no included in the encoded dataset. It is worth mentioning that once the encoded dataset is obtained, this can be manipulated in the information space by ML/statistical tasks. For obtaining interpretations for resulting models, each value in the information space can be traced to its original space by the implementation.

## Figures and Tables

**Figure 1 entropy-22-01391-f001:**
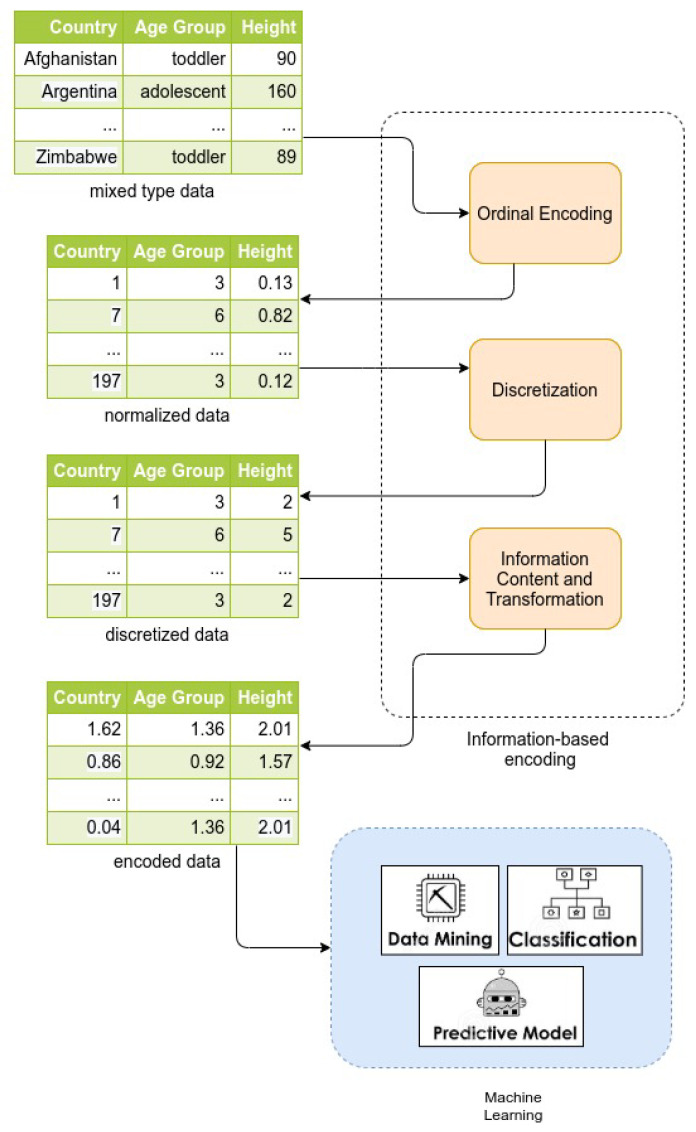
Pipeline of the proposed method for encoding a dataset.

**Figure 2 entropy-22-01391-f002:**
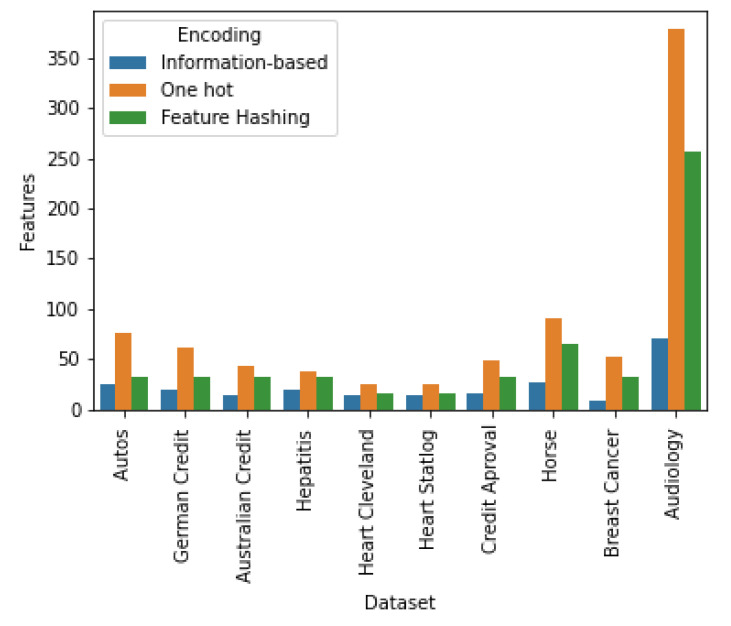
Number of features after executing the encoding process. A dataset encoded via the *information-based encoding* preserves the same number of features as the original dataset.

**Figure 3 entropy-22-01391-f003:**
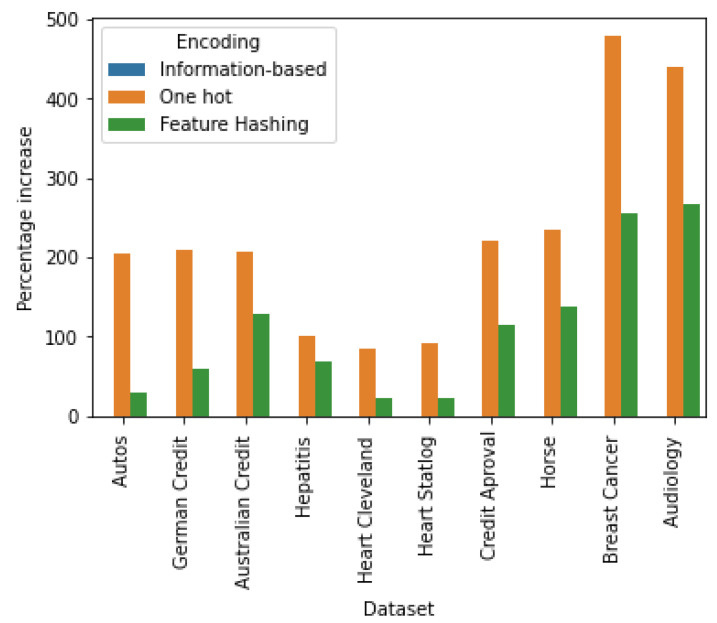
Percentage increase of the number of features. Since the *information-based encoding* does not increase the number of features, this percentage is zero.

**Figure 4 entropy-22-01391-f004:**
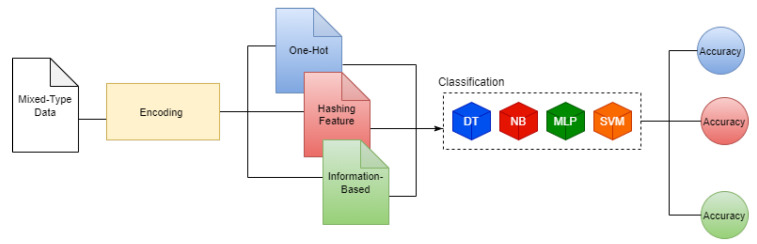
Classification Ability of Encoding Process.

**Figure 5 entropy-22-01391-f005:**
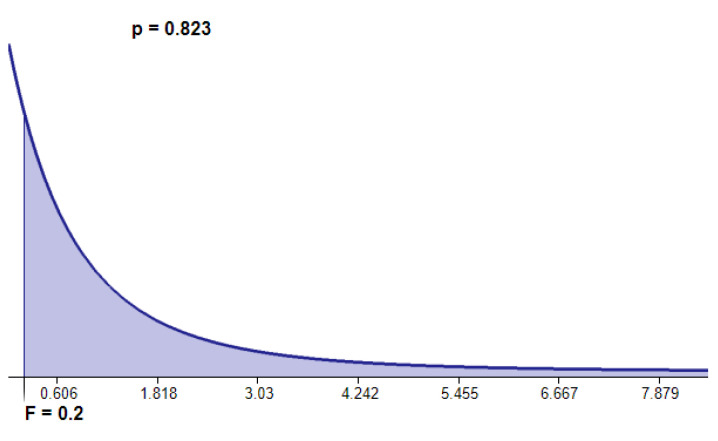
The distribution curve displays the likelihood of *F*-values for three group means. The region shaded corresponds to *F*-values greater than or equal to 0.20. When the null hypothesis is true, *F*-values fall in this area approximately 82% of the time.

**Figure 6 entropy-22-01391-f006:**
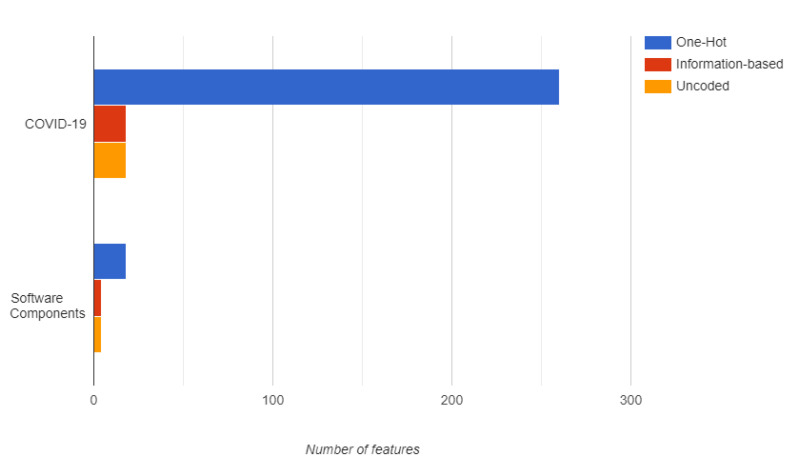
Memory efficiency in terms of number of resulting features. As reference, the number of features of the original dataset (uncoded) was included.

**Table 1 entropy-22-01391-t001:** Four categorical values encoded via one-hot scheme.

	$V_{1}$	$V_{2}$	$V_{3}$
Red	0	0	0
Green	1	0	0
Yellow	0	1	0
Blue	0	0	1

**Table 2 entropy-22-01391-t002:** Summary of feature types.

	Nominal	Ordinal	Interval	Ratio
**Properties**				
Meaningful order		✓	✓	✓
Meaningful distance			✓	✓
Absolute zero				✓
**Statistics**				
Mode	✓	✓	✓	✓
Median		✓	✓	✓
Mean			✓	✓
**Arithmetic operations**				
Addition, subtraction			✓	✓
Multiplication, division				✓

**Table 3 entropy-22-01391-t003:** Mixed-type datasets for experimentation.

Dataset	Features	Numerical	Categorical	Instances	Classes
Autos	25	15	10	205	7
German Credit	20	6	14	1000	2
Australian Credit	14	6	8	690	2
Hepatitis	19	6	13	155	2
Heart Cleveland	13	7	6	303	5
Heart Statlog	13	6	7	270	2
Credit Approval	15	6	9	690	2
Horse	27	7	20	368	2
Breast Cancer	9	0	9	201	2
Audiology	70	0	70	226	23

**Table 4 entropy-22-01391-t004:** True accuracy using different encoding approaches. This value was estimated by means of a *k*-fold cross-validation that involved a total of 1200 experiments.

Dataset	Encoding	DT	NB	MLP	SVM
	One-hot	92.10	63.62	91.06	81.66
Autos	Hashing	86.57	59.80	85.59	76.76
	Information-based	90.72	62.66	89.69	80.43
	One-hot	72.85	50.32	72.02	64.59
German	Hashing	68.48	47.30	67.70	60.71
	Information-based	73.94	51.07	73.10	65.56
	One-hot	80.55	55.64	79.64	71.42
Australian	Hashing	75.72	52.30	74.86	67.13
	Information-based	79.34	54.80	78.44	70.34
	One-hot	81.73	56.45	80.80	72.46
Hepatitis	Hashing	76.83	53.07	75.95	68.11
	Information-based	82.96	57.30	82.02	73.55
	One-hot	74.21	51.26	73.37	65.79
Heart Cleveland	Hashing	69.76	48.18	68.97	61.85
	Information-based	75.32	52.03	74.47	66.78
	One-hot	67.86	46.87	67.09	60.16
Heart Startlog	Hashing	63.79	44.06	63.07	56.55
	Information-based	66.84	46.17	66.08	59.26
	One-hot	70.28	48.54	69.48	62.31
Credit	Hashing	66.06	45.63	65.31	58.57
	Information-based	71.33	49.27	70.53	63.24
	One-hot	68.98	47.65	68.20	61.16
Horse	Hashing	64.84	44.79	64.11	57.49
	Information-based	67.95	46.93	67.17	60.24
	One-hot	96.89	66.92	95.79	85.90
Breast Cancer	Hashing	91.08	62.91	90.04	80.75
	Information-based	98.34	67.93	97.23	87.19
	One-hot	73.40	50.70	72.57	65.08
Audiology	Hashing	69.00	47.66	68.21	61.17
	Information-based	72.30	49.94	71.48	64.10

**Table 5 entropy-22-01391-t005:** Average ARI using different encoding approaches.

Dataset	Encoding	KMS	DBS	AGG
	One-hot	0.110	0.044	0.099
Autos	Hashing	0.105	0.042	0.094
	Information-based	0.112	0.045	0.101
	One-hot	−0.005	−0.002	−0.005
German	Hashing	−0.005	−0.002	−0.004
	Information−based	−0.005	−0.002	−0.004
	One-hot	0.500	0.200	0.450
Australian	Hashing	0.475	0.190	0.428
	Information-based	0.510	0.204	0.459
	One-hot	0.150	0.060	0.135
Hepatitis	Hashing	0.143	0.057	0.128
	Information-based	0.147	0.059	0.132
	One-hot	0.220	0.088	0.198
Heart Cleveland	Hashing	0.209	0.084	0.188
	Information-based	0.224	0.090	0.202
	One-hot	0.340	0.136	0.306
Heart Startlog	Hashing	0.323	0.129	0.291
	Information-based	0.333	0.133	0.300
	One-hot	0.050	0.020	0.045
Credit	Hashing	0.048	0.019	0.043
	Information-based	0.051	0.020	0.046
	One-hot	0.014	0.006	0.013
Horse	Hashing	0.013	0.005	0.012
	Information-based	0.014	0.006	0.013
	One-hot	−0.002	−0.001	−0.002
Breast Cancer	Hashing	−0.002	−0.001	−0.002
	Information-based	−0.002	−0.001	−0.002
	One-hot	0.110	0.044	0.099
Audiology	Hashing	0.105	0.042	0.094
	Information-based	0.108	0.043	0.097

**Table 6 entropy-22-01391-t006:** Real-world datasets.

Dataset	Features	Numerical	Categorical	Instances	Type of Problem
Covid-19	18	1	17	5263	Classification
Software Components	4	1	3	1407	Regression

**Table 7 entropy-22-01391-t007:** Accuracy of the model inferred via DT and MLP using the encoded versions of COVID-19 dataset. The absolute difference among accuracy values was included as indicator of discrepancy between *one-hot* and *information-based encoding*.

Fold	Classifier	Information-Based	One-Hot	Difference
1	DT	0.545	0.586	0.042
	MLP	0.713	0.617	0.097
2	DT	0.628	0.676	0.047
	MLP	0.639	0.693	0.053
3	DT	0.548	0.503	0.046
	MLP	0.400	0.535	0.135
4	DT	0.500	0.549	0.049
	MLP	0.492	0.549	0.057
5	DT	0.631	0.675	0.044
	MLP	0.688	0.751	0.063
6	DT	0.618	0.684	0.067
	MLP	0.774	0.715	0.059
7	DT	0.618	0.696	0.078
	MLP	0.760	0.715	0.046
8	DT	0.707	0.741	0.034
	MLP	0.817	0.732	0.086
9	DT	0.713	0.745	0.032
	MLP	0.808	0.766	0.042
10	DT	0.686	0.696	0.010
	MLP	0.764	0.738	0.027
			Average	0.056

**Table 8 entropy-22-01391-t008:** MSE of the model inferred via DT and MLP using the encoded versions of the software components dataset. The absolute difference among MSE values was included as indicator of discrepancy between *one-hot* and *information-based encoding*.

Fold	Regressor	Information-Based	One-Hot	Difference
1	DT	0.035	0.130	0.095
	MLP	0.044	0.099	0.055
2	DT	0.187	0.255	0.068
	MLP	0.149	0.146	0.003
3	DT	0.033	0.040	0.007
	MLP	0.059	0.043	0.016
4	DT	0.028	0.049	0.021
	MLP	0.054	0.055	0.001
5	DT	0.055	0.042	0.013
	MLP	0.070	0.044	0.026
6	DT	0.070	0.044	0.026
	MLP	0.087	0.050	0.037
7	DT	0.042	0.045	0.003
	MLP	0.064	0.046	0.018
8	DT	0.042	0.040	0.003
	MLP	0.060	0.040	0.021
9	DT	0.035	0.048	0.012
	MLP	0.079	0.036	0.042
10	DT	0.047	0.040	0.007
	MLP	0.063	0.037	0.026
			Average	0.025

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
