# Peer review of "A Memory-Efficient Encoding Method for Processing Mixed-Type Data on Machine Learning"

_entropy, 2020, doi:10.3390/e22121391_

Round 1

Reviewer 1 Report

It seems to me that the proposed method can work well when the method has knowledge from the whole examined dataset e.g. clustering case.

In classification and regression problems, the calculated values (such as min, max) based on the training set cannot be same in the testing set. The authors should comment on it.

An illustrated example in a 2d problem could help the reader to better understand the proposed method.

A statistical test should be used for the comparison of the examined methods.
The authors should present some information about the time efficiency of their method.

Author Response

Response to Reviewer 1 Comments

ROUND 1

Comment #1: It seems to me that the proposed method can work well when the method has knowledge from the whole examined dataset e.g. clustering case.

Response: The authors would like to thank the reviewer for his/her overall positive assessment of our manuscript. The reviewer has clearly pointed out what we also consider are the main strengths and contributions of our research. We have carefully reviewed the manuscript as suggested by the reviewer.

In regard to this comment, since our proposal is a preprocessing method, regardless of the ML/statistical tasks, it is mandatory to have knowledge from the whole dataset, in order to map the original values into encoded data in terms of information content. For testing purposes, the method was applied on several learning tasks. But, for demonstration purpose, in the submission, classification, regression, and clustering tasks were described. The method can be applied in similar-nature tasks.

Comment #2: In classification and regression problems, the calculated values (such as min, max) based on the training set cannot be the same in the testing set. The authors should comment on it.

Response: The reviewer is right, the text was modified for clarifying. In our work, the considered data in both the training and testing sets are not the same. In order to provide a more detailed description of the training and validation data used in our experiments, in the revised manuscript (subsection 4.2. Classification Ability), we added the following paragraph:

“In the testing stage, we make use of unknown instances (not used in the training stage). However, keep in mind that such unknown instances should be statistically similar to those used in the training stage. Furthermore, it is important to note that for both, training and testing stages, numerical and categorical values of instances have been mapped to its corresponding interval (the new numerical space -information space-) by the proposed method, "raw" numerical and categorical values are not used.”

Comment #3: An illustrated example in a 2D problem could help the reader to better understand the proposed method.

Response: For a better understanding, at the end of subsection 3.2. Information-based Encoding, we added the idea behind of the proposed method by illustrating with Figure 1. In particular, the encoded data obtained from after applying the proposed method and its usage in different machine learning tasks is now illustrated in the figure, as suggested by the reviewer.

Comment #4: A statistical test should be used for the comparison of the examined methods.

Response: We have included an Analysis of Variance in order to assess if the encoding approaches (covered in our work) have similar effects on classification and clustering. The results of this analysis were added to the end of subsections 4.2 and 4.3.

Comment #5: The authors should present some information about the time efficiency of their method.

Response: We have included an asymptotic analysis to characterize the complexity in time of our algorithm. This analysis was added to the first paragraph of subsection 3.3 Algorithm:

"Since the dataset X is composed of N d-tuples, we can see that the number of operations required by the main processes, in Algorithm 1, can be expressed as a function τ(d,N). Since typically d <= N, we could define the worst case as the one in which d=N in which case τ will grow asymptotically no faster than N2 and thus, τ(d, N) ∈ O(N2)."

Reviewer 2 Report

The paper addresses the problem of combining input variables of different nature when performing machine learning. It is really still an open problem.

The paper is clear, well organized and proposes a techniques that demonstrates good behavior and minimization of memory resources.

The method proposed is clearly explained and the experimentation is  complete and extensive.

Some comments to improve the paper:

  • The abstract has some too long sentences, which are difficult to read
  • In related work, maybe you can comment the following work (in line with Gower approach): Karina Gibert, R.Nonell,
    Impact of Mixed Metrics on Clustering. CIARP 2003: 464-471
  • In equation (8), the second case only needs to add 1 to delta if the quantity |max-min| is odd. Can you check?
  • Can you explain the criterion for choosing the datasets in the experiments (table 3)?
  • In line 314, I think 'r' shoud be 'k'.
  • In Tables 7 and 8, the performance of two versions is compared, and the difference is indicated. Can you calculate the statistical significance of that difference (maybe with an ANOVA test).
  • In the final section, the authors should discuss the limitations or drawbacks of their methods, for example: (1) they had to discard variables with equal probability on the intervals, (2) the transformation changes completely the information analysed by the ML method, so posterior interpretability may be impossible.

Minor language issues:

  • UCI reservoir -> UCI repository
  • I think the verb "resort" is not properly used
  • Line 160: feature hashing reduce ...
  • Line 200: Although a categorical feature is not strictly ...
  • Line 266: is summarized by means of Algorithm 1.

Author Response

Response to Reviewer 2 Comments

ROUND 1

Comment #1: The paper addresses the problem of combining input variables of different nature when performing machine learning. It is really still an open problem. The paper is clear, well organized and proposes techniques that demonstrate good behavior and minimization of memory resources. The method proposed is clearly explained and the experimentation is complete and extensive.

Response: The authors would like to thank the reviewer for his/her overall positive assessment of our manuscript. The reviewer has clearly pointed out what we also consider are the main strengths and contributions of our research. We have carefully reviewed the manuscript as suggested by the reviewer.

Comment #2: The abstract has some too long sentences, which are difficult to read

Response: Done, we have modified the abstract for a better reading.

Comment #3: In related work, maybe you can comment on the following work (in line with Gower approach): Karina Gibert, R.Nonell, Impact of Mixed Metrics on Clustering. CIARP 2003: 464-471.

Response: A review of this (reference [46]) and other related papers (reference [47]) by the same authors were made and included in the Related Work subsection.

Comment #4: In equation (8), the second case only needs to add 1 to delta if the quantity |max-min| is odd. Can you check?

Response: This was a typo during the edition of the equation, the "+1" was added by mistake. Also the notation of intervals had a mistake. Now both mistakes were corrected. The equation means the interval is closed at right when r is not equal to n, the interval is open at right otherwise.

Comment #5: Can you explain the criterion for choosing the datasets in the experiments (table 3)?

Response: For a better understanding, more explanation about the datasets was added in section 4. Experimental Results:

"For testing our approach and demonstrating its performance, we required mixed data type datasets, which were obtained from the UCI repository. These datasets were chosen on the assumption that was obtained from different real-world problems containing both numerical and categorical values. We wanted to test with few (9) and several (70) attributes. We included two purely categorical datasets for observing the performance of our approach with this characteristic. The main characteristic of these datasets is they have the response variable, which make them able for prediction (classification and regression) and clustering (as “ground truth” guiding).  The properties that were taken into consideration are the number of features, the number of categorical features, and the number of instances per category; the number of classes was not relevant for experiments. The experiments were carried out for demonstrating the "new" datasets can be able for ML/statistical tasks. In Table 3 a brief description of the datasets is given."

Comment #6: In line 314, I think 'r' should be 'k'.

Response: Done, we have modified the letter of the parameter.

Comment #7: In Tables 7 and 8, the performance of two versions is compared, and the difference is indicated. Can you calculate the statistical significance of that difference (maybe with an ANOVA test).

Response: We have included an Analysis of Variance (ANOVA) to test the hypothesis that the different encoded versions of the dataset induce a similar performance in ML tasks. This analysis allows us to favor the acceptance of the hypothesis suggesting that our proposal is as good as one of the most popular encodings, in terms of information, though remarkably superior in terms of memory.  The ANOVA was done to test both the classification and clustering ability. The results were added to the end of subsections 4.2 and 4.3 respectively.

Comment #8: In the final section, the authors should discuss the limitations or drawbacks of their methods, for example: (1) they had to discard variables with equal probability on the intervals, (2) the transformation changes completely the information analysed by the ML method, so posterior interpretability may be impossible.

Response: We have added an explanation of these points in a new subsection, 4.5 Discussion.

Comment #9: Minor language issues:

UCI reservoir -> UCI repository

I think the verb "resort" is not properly used

Line 160: feature hashing reduce …

Line 200: Although a categorical feature is not strictly …

Line 266: is summarized by means of Algorithm 1.

Response: Done, we have corrected the mistakes.

Round 2

Reviewer 1 Report

In the conclusion, the authors should mention the limitations of their study and how these could be handled in a future work.

Author Response

Comment #1: In the conclusion, the authors should mention the limitations of their study and how these could be handled in a future work.

Response: We have added two final remarks in the conclusions:

"As a result of encoding, some features may have identical encoded values; this is due to equiprobable values on features during the transformation process. For avoiding biased results on ML/statistical tasks, these features are no included in the encoded dataset. It is worth mentioning that once the encoded dataset is obtained, this can be manipulated in the information space by ML/statistical tasks. For obtaining interpretations for resulting models, each value in the information space can be traced to its original space by the implementation."

Also, we also make minimal changes in the conclusions.

These changes are highlighted in the modified version of our manuscript.